# Collusion of Reinforcement Learning-based Pricing Algorithms in Episodic Markets

## Abstract

Pricing algorithms have demonstrated the capability to learn tacit collusion that is largely unaddressed by current regulations. Their adoption in markets, including oligopolies with a history of collusion, necessitates further scrutiny by competition regulators. We extend the analysis of tacit collusion emerging through learning from simple pricing games to market domains that model goods with a sell-by date and fixed supply, such as airline tickets, perishables, or hotel rooms. We formalize collusion in this framework and define a measure based on the price levels under the competitive (Nash) and collusive (monopoly) equilibria. Since no analytical formulas for these prices exist, we illustrate an efficient computational method. Our experiments show that deep reinforcement learning agents learn to compete in both simple pricing games and our domain, while they show some evidence of learned collusion that warrants further analysis.

## 1. Introduction

Algorithms increasingly replace humans in pricing decisions, improving revenue and better managing complex dynamics in large-scale markets such as retail and airline ticketing. Pricing algorithms, programmed or self-learning, can engage in tacit collusion—setting *supra-competitive* prices (i.e., above the competitive level) or limiting production without explicit agreements—which eludes detection and often falls outside current competition laws (Calvano et al., 2020a). This form of collusion, generally considered illegal, undermines consumer welfare and competition. The threat is recognized by regulators, as seen in lawsuits against companies like Amazon (Bartz et al., 2023) and RealPage (Scarcella, 2023), with studies like one in Germany showing a 38% increase in fuel retailer margins post-adoption of algorithmic pricing (Assad et al., 2024). Concerns are mount-

ing among regulators (Ohlhausen, 2017; Bundeskartellamt & Autorité de la Concurrence, 2019; Directorate-General for Competition (European Commission) et al., 2019) and scholars (Harrington, 2018; Beneke & Mackenrodt, 2021; Brero et al., 2022) that *AI-based* pricing algorithms could circumvent competition laws by *learning to collude tacitly*, employing strategies unseen in (human) markets and unpredictable by (human) regulators, without illegal direct communication.

Recent research has shown that *reinforcement learning (RL)* agents can tacitly collude in simple pricing games (Calvano et al., 2020b). We extend this analysis to the new domain of *episodic markets with inventory constraints*, which model sales of perishable goods, hotel rooms, and airline tickets. We discuss the more complicated competitive and collusive equilibria of this market and present numerical methods for deriving them.

Our primary focus is airline revenue management (ARM), a market under regulatory scrutiny (European Union, 2019) with evidence of tacit collusion even before the advent of algorithmic pricing (Borenstein & Rose, 1994). With $800B in annual revenue and razor-thin net margins, this highly competitive market, regulated only by general anti-competition statutes (European Union, 2012)(Art. 101-109), has moved towards algorithmic pricing (Koenigsberg et al., 2004). Recent studies explore RL for revenue optimization (Razzaghi et al., 2022), citing multi-agent modeling as a critical next step. We close this research gap by modeling ARM as a *multi-agent RL (MARL)* problem, where independent agents optimize strategies through interaction (Busoniu et al., 2008). We employ *deep RL* to manage larger decision spaces and more complex dynamics, enhancing agents' abilities to coordinate (Li, 2018).

This paper is a first step to answering two crucial questions.

> *1. How, and in which instances, can pricing algorithms learn to collude in realistic markets?*

Determining this requires distinguishing collusive behavior from independent yet parallel responses to market conditions. We propose categorizing collusion into two types: Agents may learn collectively optimal behaviors individu-

[1]Anonymous Institution, Anonymous City, Anonymous Region, Anonymous Country. Correspondence to: Anonymous Author <anon.email@domain.com>.

ally during training, leading to immediate collusive behavior at an episode's start, as shown in (Calvano et al., 2020b). Alternatively, on an intra-episode timescale, agents might initially act competitively and use price signaling to converge on a common collusive strategy.

> *2. How can agents be prevented from learning collusion, or how can the effects of such collusion on consumers be mitigated?*

While research on mitigating strategies that target the training process or address real-time market collusion is limited, it is vital for informing policymakers to draft laws robust against algorithmic collusion (Brero et al., 2022).

In Section 2, we give an overview of related literature. In Section 3, we define the episodic, finite-horizon pricing problem with inventory constraints as a Markov Game, inspired by ARM, and formalize both competitive (Nash) and collusive (monopolistic) equilibrium strategies. With these, we define a measure that quantifies collusion in an observed episode. In Section 4, we discuss how our model's finite time horizon and inventory constraints changes the dynamics of collusion compared to previous work. In Section 5, we demonstrate efficient computation of the competitive Nash Equilibrium, a challenging task on its own.

## 2. Related Work

Our work is related to a line of research into competitive and collusive dynamics that emerge between reinforcement learning algorithmic pricing agents in economic games. We defer to Appendix C for a more detailed literature review.

Recent research most relevant to us focuses on the Bertrand oligopoly, where agents compete by setting prices, and uses Q-learning. The main line of research uses Bertrand competition with an infinite time horizon (Calvano et al., 2020b), with follow-up work varying the demand model (Asker et al., 2022), modeling sequential rather than simultaneous agent decisions (Klein, 2021), or an episodic setting with contexts (Eschenbaum et al., 2022). Findings reveal frequent, though not universal, collusion emergence, often explained by environmental *non-stationarity* preventing theoretical convergence guarantees. Agents consistently learn to charge supra-competitive prices, punishing deviating agents through 'price wars' before reverting to collusion. The robustness of collusion emergence to factors like agent number, market power asymmetry, and demand model changes underscores the potential risks posed by AI in pricing.

Which factors support and impede the emergence of learned collusion remain debated. (Waltman & Kaymak, 2008; Abada & Lambin, 2023) argue collusion results from agents 'locking in' on supra-competitive prices early on due to insufficiently exploring the strategy space, suggesting a de-

pendence on the choice of hyperparameters. Most studies identifying collusion used Q-learning, with others showing competitive behavior, raising questions about algorithm specificity (Sanchez-Cartas & Katsamakas, 2022). However, evidence from (Koirala & Laine, 2024) using Proximal Policy Optimization (PPO) in ridesharing markets suggests otherwise. We expand on these findings in a more realistic episodic, finite horizon market with inventory constraints using Deep and Multi-Agent RL through PPO, to manage our model's larger state spaces and dynamic environments.

## 3. Preliminaries and Problem Statement

### 3.1. Markov game model

We introduce a multi-agent market model for inventory-constrained goods with a sell-by date, such as perishable items, hotel rooms, or tickets, using airline revenue management (ARM) as an example. Here, agents, representing airlines, compete to sell tickets, each offering a direct flight (*single-leg*) between the same two points on the same date. This market is modeled as an episodic *Markov game* $(\mathcal{S}, \mathcal{A}, P, R, T)$ with $n$ agents (Littman, 1994). Tickets are sold over an episode with a *finite time horizon*, $t = 0, \ldots, T < \infty$. Each agent has a finite *capacity* $I_i \in \mathbb{N}$ of total seats that they can sell throughout the episode and at each time $t$, a remaining *inventory* of tickets $x_{i,t} \in \{0, \ldots, I_i\}$, resulting in an inventory vector $x_t = (x_{1,t}, \ldots, x_{n,t})$. An agent's *marginal cost* per sold ticket, $c_i$, is constant.

Each period, all agents observe the current state $s_t \in \mathcal{S}$ and each simultaneously use their *policy* $\pi_i : \mathcal{S} \to A_i$ to choose an *action* in the form of a price $p_{i,t} = \pi_i(s_t)$, forming the price vector $p_t = (p_{1,t}, \ldots, p_{n,t})$. The state $s_t = (p_{t-1}, x_t)$ comprises the last price and inventory vectors, representing a one-period memory. We also assume *full observability*, allowing all agents to see competitors' past prices and current inventories. Real-time information on offered ticket prices and inventories (though airlines may hold some in reserve) is collected by *Global Distribution Systems (GDS)* like Amadeus, and is publicly available at a cost.

State transitions occur according to $P(s_{t+1}|s_t, p_t)$. For each agent $i$, the market determines a *demand* $d_{i,t}$ at time $t$, the agent sells a corresponding *quantity* $q_{i,t} = \min(d_{i,t}, x_{i,t})$ and their inventory is updated to $x_{i,t+1} = x_{i,t} - q_{i,t}$. Finally, the agent receives a *reward* corresponding to their profit $R_{i,t} = (p_{i,t} - c_i)q_{i,t}$. Agents pick actions aiming to maximize expected future rewards $\mathbb{E}[\Sigma_{s \geq t}^T R_{i,s}]$. The initial state $s_0$ uses dummy values, signaling the beginning of an episode and allowing agents to choose the initial prices. To obtain a finite action space necessary for many learning algorithms, we model it as a discretized interval of possible prices. We do not model cancellation and overbooking.

### 3.2. Demand model

We employ a modified *multinomial logit (MNL)* demand model, commonly used in Bertrand price competition, to simulate the probability of a customer choosing each agent's product, ensuring demand distribution among all agents rather than clustering on the "best" offering.

Each agent's product has a quality $\alpha_i$. There is an outside good with quality $\alpha_0$ for vertical differentiation, and a parameter $\mu$ that signifies horizontal differentiation. The *demand* for product $i = 1, \ldots, n$ in period $t$ is given by $d(p_{i,t}, p_{-i,t}) := \lfloor \lambda d_{i,t} \rfloor$, where

$$d_{i,t} = \frac{\exp\big((\alpha_i - p_{i,t})/\mu\big)}{\sum_{j \in N_t^a} \exp\big((\alpha_j - p_{j,t})/\mu\big) + \exp(\alpha_0/\mu)} \in (0,1),$$

$N_t^a := \{j \in N \mid x_{j,t} > 0\}$ and $\lambda \in \mathbb{N}$.

We incorporate *choice substitution*, or *demand adaptation*, by summing only over agents with available inventory. If an agent is sold out, demand shifts to those with remaining inventory, preventing the sold-out agent's actions from affecting the demand and rewards of others. Demand values are scaled by $\lambda > 1$ and rounded to the nearest integer to account for the sale of goods (tickets) in whole numbers.

### 3.3. Measuring collusion and competition

We categorize an observed episode and observed agent strategies on a scale from "competitive" to "collusive" by defining an episodic collusion measure. We first define the two necessary equilibria in the Markov game:

**Definition 3.1.** A collection of agent policies is called

- Competitive, or *Nash equilibrium* if no agent $i$ can improve their expected total episode profit $\mathbb{E}[\Sigma_{t=1}^T R_{i,t}]$ by unilaterally picking a different policy given fixed opponent strategies.
- Collusive, or *monopolistic equilibrium* if it maximizes the expected *total* agent profit $\mathbb{E}[\Sigma_{i=1}^n \Sigma_{t=1}^T R_{i,t}]$.

Using the prices set by agents in both the Nash- and monopolistic equilibrium, $p^N$, and $p^M$, and the corresponding agent profits $R^N, R^M$, we can define the following measure.

**Definition 3.2.** For agent $i$'s profit in the observation, Nash, and collusive equilibria as $\bar{R}_i, R_i^N, R_i^M$ respectively, the agent's *episodic profit gain* is

$$\Delta_{i,e} := \frac{1}{T} \sum_{t=1}^T \frac{\bar{R}_{i,t} - R_{i,t}^N}{R_{i,t}^M - R_{i,t}^N}.$$

The *episodic collusion index* is calculated as

$$\Delta_e := \Big(\prod_{i=1}^n \Delta_{i,e}\Big)^{\frac{1}{n}},$$

indicating a competitive or collusive outcome at 0 or 1, respectively.

We employ the geometric mean in our collusion index, as opposed to the simple average used in previous studies (Calvano et al., 2020b; Eschenbaum et al., 2022), as it more strongly penalizes unilateral competitive defections in a collusive arrangement. Exploring alternative measures, which could be inspired by social choice theory, is a promising avenue for future research.

## 4. Overview of the collusive strategy landscape

Previous work has focused on infinitely repeated games. We discuss how our model's episodic nature and finite inventory significantly affect the strategies for establishing and maintaining learned tacit collusion. In general, collusion must first be established by agents exploring non-competitively optimal behaviors and discovering mutually beneficial strategies, or using actions as covert signals to communicate and form agreements. To maintain collusive agreements, agents need to remember past actions and have mechanisms to punish those who deviate from the agreed-upon strategy.

**Infinitely repeated games** These settings allow deriving competitive and collusive equilibrium price levels through implicit formulas. They provide the most room for collusive strategies to emerge and sustain. Typically, stable collusion manifests in two forms. First, *reward-punishment schemes:* Agents cooperate by default and punish deviations. A deviating agent is punished by others charging competitive prices, thereby removing the benefits of collusion temporarily, until the supra-competitive prices are reinstated. This dynamic involves agents synchronizing over rounds to restore higher price levels after a deviation. This pattern can be observed as fixed, supra-competitive prices and verified by forcing one agent to deviate and recording everyone else's responses, as done in (Calvano et al., 2020b). Second, *Edgeworth price cycles:* This pattern involves agents sequentially undercutting each other's prices until one reverts to the collusive price, prompting others to follow, restarting the undercutting cycle (Klein, 2021).

**Episodic games** Collusive strategies can now emerge either *intra-episode* through action-based communication or *across multiple episodes*, with agents displaying collusion from the onset of a new episode. (Eschenbaum et al., 2022) find that the latter form, possibly due to strategy overfitting to familiar opponents, is prevalent in oligopolistic settings, seeing collusive agents play competitively against new opponents before re-establishing collusion through continued learning. We are especially interested in observing intra-episode collusion, as many real marketplaces feature frequently changing participants.

The episodic nature limits the efficacy of traditional reward-punishment schemes in maintaining collusion. If every single period of the game has a unique Nash equilibrium, as is the case in the Bertrand setting, backward induction from the last timestep $T$ suggests agents should deviate to play the Nash strategy, undermining stable collusion. Does this mean that collusion in episodic games is impossible? No: If agents remember past interactions across episodes, past deviations can be punished in future episodes. Our experiment in Figure 2 shows that even without that possibility, if episodes are long enough, learning agents may still converge to collusive strategies of the signaling, stable or cyclic kind, as discovering the backward induction argument through (often random) exploration may be unlikely enough in practice.

**Our model** Besides the episodic structure, inventory constraints significantly expand the state and strategy space by linking pricing to inventory levels, complicating the prediction and interpretation of collusion. Determining the competitive and collusive price levels becomes more complex because the solution formulas from the Bertrand or Cournot settings require smoothness or convexity assumptions that no longer hold. We approach finding a Nash equilibrium by modeling each episode as a simultaneous-move game where agents set entire price vectors, detailed in Section 5.1. We solve resulting generalized Nash equilibrium problem numerically and prove that its solutions of are Nash equilibria in our Markov game. We find that in our model, collusion can occur without a punishment scheme: given fixed total demand and sufficient (surprisingly light) inventory constraints, competitive pricing may naturally align with collusive levels. We see that both episodic equilibria consist of repeating their one-period equivalents $T$ times. If agents discount future rewards, both equilibria shift to lower prices and higher profits early in the episode and vice-versa toward its end. Additionally, the competitive and collusive equilibria remain distinct even with strict inventory constraints. Due to the difficulty of predicting or interpreting observed behavior in this complex setting, we see value in analyzing different types of learners as part of future work.

## 5. Experiments

Our experiments explore obtaining Nash and collusive equilibria in our episodic market model. We present initial results from settings with and without inventory constraints, where a learner exhibits collusion in the episodic setting and competition when inventory constraints are present.

### 5.1. Obtaining competitive and collusive price levels

Previous works' Bertrand settings use analytic formulae to compute Nash and monopolistic equilibrium price vectors $p^N$ and $p^M$ for single-period cases. However, our multi-period model and the complexity added by inventory constraints necessitate a different approach. We model an entire episode as a *simultaneous-move game (SMG)*, where all agents $i$ must simultaneously decide all $T$ prices in their vector $p^{(i)} = (p_{i,1}, \ldots, p_{i,T})$. Let $p = (p^{(1)}, \ldots, p^{(n)})$ encompass all agents' price vectors, with $p^{(-i)}$ representing all agents' vectors except $i$'s. Each agent, given fixed opponent strategies $p^{(-i)}$, aims to solve:

$$\max_{p^{(i)}} \quad \sum_{t=1}^{T} (p_{i,t} - c) \lfloor \lambda d_{i,t} \rfloor \tag{1}$$

$$\text{subject to} \quad \sum_{t=1}^{T} \lfloor \lambda d_{i,t} \rfloor \leq I, \quad p^{(i)} \geq 0. \tag{2}$$

**Definition 5.1.** The *Generalized Nash Equilibrium Problem (GNEP)* consists of finding the price vector $p^* = (p^{(1)*}, \ldots, p^{(n)*})$ such that for each agent $i$, given $p^{(-i)*}$, the vector $p^{(i)*}$ solves their inventory-constrained revenue maximization problem.

This vector represents a Nash equilibrium, as each agent maximizes their revenue under the assumption of fixed competitor actions. Solving the GNEP is difficult since each agent's constraints depend on the other agents' strategies through the MNL demand $d_{i,t}$, which is a function of both agent $i$'s and the other agents' chosen prices. A solution price vector can be interpreted as the *actions* of a set of (unknown) agent policies playing an episode of the Markov game. Above we assumed that environment transitions and initial state are deterministic.

**Lemma 5.2.** *Assume deterministic transitions and policies playing pure strategies. Let $p^* = (p^{(1)*}, \ldots, p^{(n)*})$ from the SMG solve the GNEP. Then, the set of policies $\pi^* = (\pi_1^*, \ldots, \pi_n^*)$, where $\pi_i^*(s_t) = p_{i,t}^*$ for all $i$, $t$, and $s_t \in \mathcal{S}$, is a Nash equilibrium in the Markov Game.*

The full proof can be found in Appendix D. Details on our numerical approach for solving the GNEP are found in Appendix A.

We find that in the undiscounted case, the episodic equilibrium price vectors repeat the single-period equilibrium with the same parameters $T$ times. Figure 3 illustrates how inventory constraints influence the market's competitive dynamics. If agents' inventories are bigger than the demand they would satisfy in the competitive equilibrium, the equilibria correspond to the unconstrained setting. As inventories are set smaller, the competitive price level increases as it becomes harder for firms to undercut and profit from the increased demand. At a constraint level equaling the demand under the collusive price, there is no more room for competition and tightening constraints even further increases the now coinciding competitive and collusive prices. We set inventory constraints on agents to a level between

the demands at monopolistic and Nash equilibrium prices, allowing differentiation between competitive and collusive behavior and a well-defined collusion index.

### 5.2. Learned collusion in our model

We implement our agents using proximal policy optimization (PPO), contrasting with prior research that predominantly employs tabular Q-learning in the Bertrand setting. Given the complex state space of our model, function approximation is necessary, and the choice of PPO over deep Q-networks (DQN) adds to the discussion of how collusion emergence is influenced by the learning algorithm used. PPO operates by simulating trajectories and adjusting the policy distribution's parameters based on observed outcomes. To discover collusion, agents must initially explore sufficiently but reduce exploration once collusion is established to avoid random deviations that might disrupt the collusive agreement. Since PPO picks actions via sampling from its policy distribution, controlling the degree of exploration vs exploitation is not as straight-forward as tuning the previously used $\epsilon$-greedy (deep) Q-learners' parameter $\epsilon$. We implement PPO and the market environment using JAX, with training logic adapted from (Willi et al., 2023).

Our evaluation setup features two agents and a five-step time-horizon. The other parameters are inspired by (Calvano et al., 2020b) and can be found in Appendix B.

Figures 1 and 2 demonstrate that PPO agents can learn to set supra-competitive prices in non-inventory constrained episodic settings. This behavior hinges on training over numerous epochs (50) on single-episode rollouts. Larger learning rates disrupt this collusion, aligning with previous works' findings that used Q-learners. One can guide PPO agents toward quickly learning competition in our market environment by using rollouts with a large amount (e.g., 4096) of episodes between training steps, simulated in parallel. This works with learning rates as small as 0.0003 or as large as 0.01 and an entropy coefficient of 0.01.

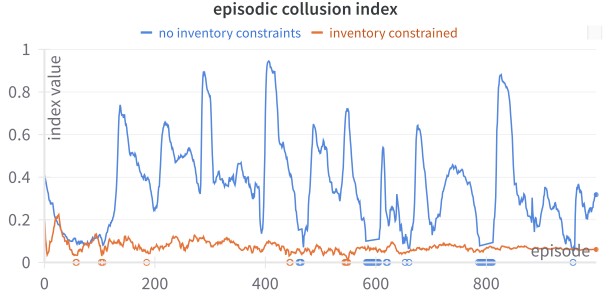

*Figure 1.* With constrained inventory, agents learn competition. Without inventory constraints, they display cyclic collusion.

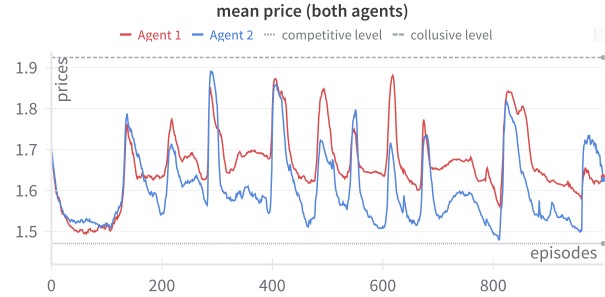

*Figure 2.* In an episodic, non-inventory constrained setting, agents display cyclic supra-competitive prices.

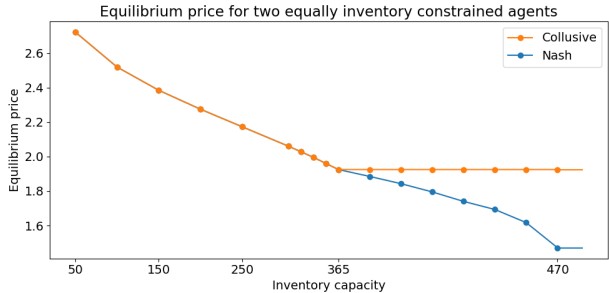

*Figure 3.* The effect of inventory constraints on the one-period equilibrium price levels for two agents with equal inventory capacities. The demand for each agent in the competitive and collusive equilibrium is 470 and 365, respectively.

## 6. Conclusion

We have developed a Markov game model tailored for Airline Revenue Management (ARM), facilitating the analysis of tacit collusion within finite time horizon and inventory-constrained markets. We have shown methods to obtain competitive and collusive equilibria in our model. Additionally, we have deployed a multi-agent reinforcement learning framework using proximal policy optimization (PPO), showing that agents can both learn to compete in our model as well as engage in collusive behavior if inventory constraints are lifted.

Future efforts will focus on a deeper exploration of the potential for MARL algorithms like PPO and opponent-shaping agents (Souly et al., 2023) to facilitate collusion. We aim to develop strategies to prevent collusion from being learned in training (Brero et al., 2022) or established through real-time market signaling. We plan to enhance our model with additional ARM-specific elements such as overbooking and cancellation policies.

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

## A. Numerical solution strategy for Nash and monopolistic equilibria

To solve the GNEP for competitive equilibrium prices, we use a Gauss-Seidel-type iterative method (Facchinei & Kanzow, 2007). We start with an initial price vector guess and proceed through a loop where each iteration updates each agent's price by solving their subproblem. For agent $i$ at iteration $k$, it uses the fixed opponent prices from the latest estimate. The process repeats until convergence to $p^*$.

Each agent's subproblem is a mixed-integer, nonlinear optimization problem (MINLP), with neither convex objectives nor constraints. We use *Bonmin*, a local solver capable of handling larger instances at the risk of missing global optima. We mitigate this by initiating the solver from multiple different starting points. For the collusive equilibrium, we simulate a scenario where one agent sells $n$ items, aiming to maximize the total episodic revenue under $n$ inventory constraints. This problem is again a non-convex MINLP. Our implementation uses the open-source COIN-OR solvers via Pyomo in Python.

## B. Evaluation setting

Our evaluation setting features $n = 2$ agents, qualities $\alpha_i = 2$, equal marginal costs $c = c_i = 1 \ \forall i$, a horizontal differentiation factor of $\mu = 0.25$, an outside good quality of $\alpha_0 = 0$, demand scaling factor of $\lambda = 1000$ and inventory constraints of $420 * 5 = 2100$. The prices and demands in the unconstrained one-period Nash and monopolistic equilibria are $p^N = 1.471, p^M = 1.925$ and $d^N = 470, d^M = 365$ respectively. The constrained case features the identical monopolistic equilibrium, but a Nash equilibrium with $p^N = 1.759$ and $d^N = 420$. Agents choose prices from a discretized interval $[p^N - \xi(p^M - p^N), p^M + \xi(p^M - p^N)]$ with 20 steps and $\xi = 0.231$.

## C. Literature Review

**Examples and description of tacit collusion**   Firms across various sectors, from insurance to flight tickets, employ *algorithmic pricing* to maximize revenue by leveraging data on market conditions, customer profiles, and other factors. These algorithms' growing complexity raises challenges for maintaining fair competition and detect firms that *tacitly collude*, ones which jointly set *supra-competitive* prices (i.e., above the competitive level) or limit production *without explicit agreements or communication*. Recently, evidence has emerged that companies are already using algorithmic pricing to inflate prices market-wide at the cost of consumers. For instance, (Assad et al., 2024) showed that German fuel retailer margins increased by 38% following the widespread adoption of algorithmic pricing. Other examples are found in setting credit card interest rates (Ausubel, 1991) and consumer goods markets (Genesove & Mullin, 2001).

**Legal developments around algorithmic collusion**   Current anti-collusion policies mainly address explicit agreements, making tacit collusion, which is inferred from company behaviors rather than evidence of an agreement, more elusive and difficult to prove. There is growing concern among regulators (Ohlhausen, 2017; Bundeskartellamt & Autorité de la Concurrence, 2019; Directorate-General for Competition (European Commission) et al., 2019) and researchers (Harrington, 2018; Beneke & Mackenrodt, 2021; Brero et al., 2022) that AI-based pricing algorithms might evade competition laws by colluding tacitly, without direct communication or explicit instruction during learning. This highlights the need for better strategies to prevent collusion or mitigate its negative effects on the market.

**Reinforcement learning (RL) background**   *Reinforcement learning* (Sutton & Barto, 2018) is an advanced segment of machine learning where agents learn to make sequential decisions by interacting with an environment. Unlike traditional machine learning methods which rely on static datasets, RL emphasizes the development of autonomous agents that improve their behavior through trial-and-error, learning from their own experiences. This approach enables agents to understand complex patterns and make optimized decisions in scenarios with uncertain or shifting underlying dynamics. *Multi-agent* RL extends this concept to scenarios involving multiple decision-makers, each optimizing their strategies while interacting with others and the environment (Busoniu et al., 2008). In MARL settings, agents can be incentivized to behave competitively, as seen in zero-sum games like Go (Silver et al., 2017; 2018), cooperatively, like in autonomous vehicle coordination (Dinneweth et al., 2022) or a mix of the two that includes our problem, i.e., markets and pricing games. MARL, while posing challenges such as *non-stationarity* and *scalability*, enables agents to adapt to and influence competitors' strategies, facilitating tacit collusion.

**Collusion & regulation in airline revenue management (ARM)**   Originally a strictly regulated sector with price controls, ARM was deregulated in 1978 in the US and Europe, leading to a competitive landscape of private carriers whose

pricing strategies are subject only to general laws against anti-competitive behavior (European Union, 2012)(Art. 101-109). However, this deregulation has caused market consolidation, prompting regulatory responses to protect competition (European Union, 2019). Even prior to algorithmic pricing, regulators have identified pricing behaviors suggestive of tacit collusion (Borenstein & Rose, 1994), underscoring the challenge of distinguishing between collusive behavior and independent but parallel responses to market conditions.

**Background on the field of revenue management (RM)**   Each of the agents that we model is individually maximizing their revenue, relating our work to the field of *revenue management (RM)* (Talluri & Van Ryzin, 2004). As a competitive market with slim net margins, airlines are increasingly turning to *dynamic pricing* (Koenigsberg et al., 2004) beyond traditional *quantity-based* and *price-based* RM, replacing the hugely popular expected marginal seat revenue (EMSR) models (Belobaba, 1987). Our problem falls into the price-based RM category, even though we do model aspects of capacity management with our inventory constraints. In quantity-based RM, agents decide on a production quantity with the price for their good being the result of a market-wide fixed function of that decision, and models often impose no limit on the offered quantity. In our model, agents decide their price, and demand results from a market-wide function. Our aim is for agents to learn to predict the impact of their pricing choices on the demand and thus sold quantity, in order to optimally use the constrained inventory that they have.

**Learning in general RM**   In recent years, reinforcement learning agents have seen increased use in revenue management outside of the airline context. Examples include learning both pricing and production quantity strategies in a market with perishable goods (Wang et al., 2021), producing a pricing policy by learning demand (Rana & Oliveira, 2014; 2015) and analyzing the performance of different popular single-agent RL in various market settings (Kastius & Schlosser, 2022) (here Q-learning and Actor-Critic). The use of largely uninterpretable learned choice or pricing models introduces new challenges, such as deriving economic figures like the elasticity of demand with respect to price (Acuna-Agost et al., 2023).

**Learning in ARM**   While early work used e.g. heuristically solved linear programming formulations (Bront et al., 2009) or custom learning procedures (van Ryzin & McGill, 2000; Bertsimas & de Boer, 2005), recent studies have explored single-agent reinforcement learning in ARM to learn optimal pricing (Razzaghi et al., 2022). These model the problem as a single-agent Markov decision problem (MDP) (Gosavi et al., 2002; Lawhead & Gosavi, 2019) and consider various realistic features like cancellations and overbooking (Shihab & Wei, 2022). The application of *deep reinforcement learning (deep-RL)* (Mnih et al., 2015) is growing in this complex market (Bondoux et al., 2020; Alamdari & Savard, 2021), but these models often overlook the multi-agent nature of the airline market. We model the market as a multi-agent system with individual multi-agent learners, a critical yet unexplored aspect in current research (Razzaghi et al., 2022).

## D. Proof of Lemma 1

*Proof.*   Let us introduce some terminology first.

**Definition D.1.** Fix an agent $i$ with policy $\pi_i$ or price vector $p^{(i)}$, and fix opponent policies $\pi^{(-i)}$ or prices $p^{(-i)}$.

- A *useful deviation* is a policy $\pi_i'$ or price vector $p^{(i)'}$ that strictly increases $i$'s revenue over the whole episode compared to playing $\pi_i$ or $p^{(i)}$. We use this term in both the Markov game and SMG.

- We call a price vector $p^{(i)} = (p_{i,1}, \ldots, p_{i,T})$ *feasible in the GNEP* if it fulfils the inventory constraint of $i$'s revenue maximization problem in Equation (1), and *infeasible in the GNEP* if it doesn't.

- We call a policy $\pi_i$ *simple*, if at each time $t$, it outputs the same value for all states $s_t$, i.e. $\forall t \, \forall s_t : \pi_i(s_t) \equiv \text{const}_t$.

Intuitively, we construct a set of simple policies where each agent always plays their GNEP solution, no matter the state, and show that this set of policies is a Nash equilibrium.

First, observe that those simple policies result in the same set of price vectors $p^*$ in every evolution of the Markov game. In particular, fixing opponent strategies $\pi^{(-i)*}$ results in agent $i$ facing the same fixed opponent price vectors $p^{(-i)*}$ (from the GNEP solution) in every evolution of the Markov game. Therefore, to prove that $\pi^*$ is a Nash equilibrium in the Markov game it's enough to prove that for any agent $i$ and fixed opponent price vectors $p^{(-i)*}$, there doesn't exist a useful deviation price vector $p^{(i)'} \neq p^{(i)}$. If a useful deviation policy $\pi_i'$ existed for $i$, in at least one timestep $t$ it would have to pick a price $p_{i,t}' \neq p_{i,t}$, so by ruling out a useful price vector deviation we also rule out a useful policy deviation.

**Claim:** Let $p^{(-i)}$ be fixed opponent price vectors. Given any price vector $p^{(i)}$ for agent $i$, there always exists a price vector $\bar{p}^{(i)}$ that is feasible in the GNEP and such that playing $\bar{p}^{(i)}$ results in revenue for $i$ that is as great as or greater than that from playing $p^{(i)}$.

Given opponent prices $p^{(-i)*}$, if a useful deviation $p^{(i)'} \neq p^{(i)*}$ exists for agent $i$, it must be infeasible in the GNEP (otherwise $p^{(i)*}$ wouldn't be a revenue-maximizing solution to agent $i$'s GNEP's subproblem). However, since the claim implies that we could construct a $\bar{p}^{(i)}$ that is feasible in the GNEP and has equivalent revenue for $i$ as the infeasible $p^{(i)'}$, it would be a useful deviation for agent $i$ in the SMG to play $\bar{p}^{(i)}$ given $p^{(-i)*}$, contradicting the assumption that $p^*$ is a NE.

**Proof of Claim:** Let opponent prices be fixed $p^{(-i)}$. Let $p^{(i)}$ a price vector in the Markov game that's infeasible in the GNEP (otherwise we're trivially done). Let $i$'s inventory at $t$ be $x_t$. Let $\hat{t} \in \{1, \ldots, T\}$ be the *sell-out time*, i.e. the last timestep in which $i$ has nonzero inventory, meaning $\hat{t} := \max\{t \in \{1, \ldots, T\} | x_{\hat{t}} > 0\}$ such that $x_{\hat{t}} = 0$ and $\forall t > \hat{t} : x_t = 0$. Let $d(p_{i,t}, p_{(-i),t}) := \lfloor \lambda d_{i,t} \rfloor$ be the scaled, truncated MNL demand of agent $i$ at time $t$ given price vector $p$, which is a decreasing function in $p_{i,t}$.

Define

$$\bar{p}_{i,\hat{t}} := \sup\{q \mid d(q, p_{(-i),\hat{t}}) = x_{\hat{t}}\}$$
$$\bar{p}_{i,t} \in \{q \mid d(q, p_{(-i),t}) = 0\} \quad \forall t > \hat{t}.$$

Then, let $\bar{p}^{(i)} := (p_{i,1}, \ldots, p_{i,\hat{t}-1}, \bar{p}_{i,\hat{t}}, \bar{p}_{i,\hat{t}+1}, \ldots, \bar{p}_{i,T})$.

Given the other agents' fixed price vectors $p^{(-i)}$, the vector $\bar{p}^{(i)}$ is feasible in the GNEP. To see this, consider that every price vector has a sell-out time $\hat{t}$. At any point in time before $\hat{t}$, the accumulated demand up until that time is lower than inventory, otherwise $\hat{t}$ wouldn't actually be the sell-out time. The GNEP's feasibility constraint is only violated if at $\hat{t}$, demand is larger than remaining inventory $x_{\hat{t}}$, or if at any $t > \hat{t}$, demand is larger than $0$. The construction of $\bar{p}^{(i)}$ ensures that it has the same sell-out time $\hat{t}$, and the construction of $\bar{p}_{i,t}$ for $t \geq \hat{t}$ ensures that demand at $\hat{t}$ matches inventory left, and that demand at $t > \hat{t}$ is zero, meaning that $\bar{p}^{(i)}$ cannot violate the feasibility constraint.

Now we just need to prove that given fixed opponent prices $p^{(-i)}$, agent $i$'s reward in the Markov game when playing $\bar{p}^{(i)}$ is as great as or greater than their reward when playing $p^{(i)}$. Their reward when playing $p^{(i)}$ is given by

$$\Sigma_{t=1}^{\hat{t}-1}(p_{i,t} - c) \min\left(d(p_{i,t}, p_{(-i),t}), x_t\right) + (p_{i,\hat{t}} - c) \min\left(d(p_{i,\hat{t}}, p_{(-i),\hat{t}}), x_{\hat{t}}\right) + \Sigma_{t=\hat{t}+1}^{T}(p_{i,t} - c) \min\left(d(p_{i,t}, p_{(-i),t}), x_t\right)$$

We now replace $p^{(i)}$ with $\bar{p}^{(i)}$ and compare each term.

In the *first term*, as we know that for $t < \hat{t}$ $i$'s demand is always lower than their inventory by definition of $\hat{t}$, the term reduces to
$$\Sigma_{t=1}^{\hat{t}-1}(p_{i,t} - c)d(p_{i,t}, p_{(-i),t}).$$

Since $p_t = \bar{p}_t$, we see that the first revenue term's value stays equal:

$$\Sigma_{t=1}^{\hat{t}-1}(p_{i,t} - c)\min\left(d(p_{i,t}, p_{(-i),t}), x_t\right) = \Sigma_{t=1}^{\hat{t}-1}(p_{i,t} - c)d(p_{i,t}, p_{(-i),t}) = \Sigma_{t=1}^{\hat{t}-1}(\bar{p}_{i,t} - c)d(\bar{p}_{i,t}, p_{(-i),t}).$$

In the *second term*, by definition of $\hat{t}$, we know that $\min\left(d(p_{i,\hat{t}}, p_{(-i),\hat{t}}), x_{\hat{t}}\right) = d(p_{i,\hat{t}}, p_{(-i),\hat{t}}) = x_{\hat{t}}$, thus the term reduces to

$$(p_{i,\hat{t}} - c)d(p_{i,\hat{t}}, p_{(-i),\hat{t}}).$$

Since $d(p_{i,\hat{t}}, p_{(-i),\hat{t}}) \geq x_{\hat{t}}$, and by construction $d(\bar{p}_{i,\hat{t}}, p_{(-i),\hat{t}}) = x_{\hat{t}}$, and $d(\cdot, p_{(-i),\hat{t}})$ decreasing, we get $\bar{p}_{i,\hat{t}} \geq p_{i,\hat{t}}$. We also know that $i$ will always choose a price $\geq c$ to ensure non-negative revenue. Thus, we see that the second revenue term's value can only increase:

$$(p_{i,\hat{t}} - c) \min\left(d(p_{i,\hat{t}}, p_{(-i),\hat{t}}), x_{\hat{t}}\right) = (p_{i,\hat{t}} - c)d(p_{i,\hat{t}}, p_{(-i),\hat{t}}) \leq (\bar{p}_{i,\hat{t}} - c)d(\bar{p}_{i,\hat{t}}, p_{(-i),\hat{t}}).$$

In the *third term*, by definition of $\hat{t}$, we know that $\forall t > \hat{t} : x_t = 0$, and since by construction of $\bar{p}^{(i)}$ we also know that $\forall t > \hat{t} : d(\bar{p}_{i,t}, p_{(-i),t}) = 0$, we see that the term's value remains zero:

$$\Sigma_{t=\hat{t}+1}^{T}(p_{i,t} - c) \min\left(d(p_{i,t}, p_{(-i),t}), x_t\right) = \Sigma_{t=\hat{t}+1}^{T}(\bar{p}_{i,t} - c)d(\bar{p}_{i,t}, p_{(-i),t}) = 0$$

Putting all three terms together, agent $i$'s revenue from playing $\bar{p}^{(i)}$ is as great as, or greater than that from playing $p^{(i)}$. $\square$