# OpenReview forum: "Collusion of Reinforcement Learning-based Pricing Algorithms in Episodic Markets"
_ICML.cc/2024/Workshop/Agentic_Markets — Agentic Markets @ ICML'24 Poster_

### Official Review · Reviewer_9u81 · 2024-06-12
**A very relevant problem with useful formalisms, fairly realistic market model assumptions and promising results requiring further ablations**

**Rating:** 7
**Confidence:** 4

**Review:**

The paper investigates the emergence of tacit collusion among pricing algorithms, particularly in markets with perishable goods. These algorithms can set supra-competitive prices or limit production without explicit agreements, a practice that current competition laws often fail to detect. The study extends previous research to more complex episodic markets, focusing on airline revenue management. It introduces a measure for collusion, and demonstrates efficient computational methods to identify collusive behavior in the more realistic non-infinitely repeating cases when one cannot be analytically derived. Experiments  in price prediction and collusion detection show that deep reinforcement learning agents can learn to compete and exhibit collusive tendencies. The research addresses key questions about how and when algorithms collude and how to mitigate these effects, emphasising the need for robust regulatory frameworks to handle algorithmic collusion.

Several areas in the paper leave room for improvement or further explanation. The authors use a deterministic policy as the action is taken to maximise the episodic reward and the state is defined by past price and inventory vectors. This seems a rather rigid assumption as airlines may operate under temporary promotions or targeted seasonal pricing within an episode. While this is in no way detrimental to the quality of results, it would be really interesting to see the use of a stochastic policy and what it represents in this context. Following on the RL assumptions, the inflexibility of PPO in directly managing exploration rate is mentioned without any further connection to the rest of the results. PPO does not have an epsilon parameter, but exploration can be controlled through hyperparameters like entropy coefficients or by using an action distribution in training time and only taking the mean when testing in a deterministic manner. More ablations of this kind would be in order to dermine what part of the results is related to the limited capacity and initializations of different networks.

The use of the geometric mean for the collusion index diverging from the common practice of taking the average is noted but not properly explained. It would be helpful to the reader to explain what other measures social choice theory inspires and why should this particular type of competitive defection be more strongly penalised in this work.

As pedantic as the following edit is, the future work paragraph in the Conclusion section refers to PPO as a MARL algorithm which is misleading as PPO is also a single agent training algorithm.

The paper is well-written and addresses a very relevant problem both to the state of modern markets and the current workshop. The workshop would greatly benefit from having this work encourage discussions on training collusion-avoidant agents.

---

### Official Review · Reviewer_gp4Y · 2024-06-13
**Strong technical contribution in stylised model of Airline Revenue Management**

**Rating:** 7
**Confidence:** 4

**Review:**

On the scientific content

* The paper makes several contributions, namely the novel contribution of episodic market analysis to markets with inventory constraints and perishable goods, specifically Airline Revenue Management.
* The paper's introduction (and appendix literature section) explains the necessity of investigating the problem in this domain in detail.
* The Markov game formulation with the new additions that break the common assumptions of smoothness and convexity are refreshing. The choice to model this as a simultaneous-move game is fair, though assumes full observability, for which it is unclear whether this realistically applies to this application domain.
* The methods (PPO) used are appropriate for the complicated action space
* Strong technical contributions in terms of the framework and numerical methods.
* Though the recent regulation against anti-competitive behaviour is mentioned, what are the direct policy implications of this work?
* How would the authors empirically validate their work?

On the writing
* Referencing the relevant regulation and past instances of collusion helps the reader better understand the context of the problem.
* Market power asymmetry is named in the related work section but is not investigated further or explained how this may affect the model developed here.
* Over-emphasizes previous work
* Page 3 line 135 (right column)
> These settings allow **for** deriving..
* Page 4, lines 191,192:
> We solve **the** resulting generalized Nash equilibrium problem numerically and prove that its solutions **~~of~~** are Nash equilibria in our Markov game.
* Making the source code public allows others to replicate the results